# Epidemiology, clinical features, and impact of food habits on the risk of hepatocellular carcinoma: A case-control study in Bangladesh

**M. Al-Amin Shawon[1]☯, M. Abul Khair Yousuf[2]☯, Enayetur Raheem[3]☯, Sium Ahmed[1], Tyeaba Tasnim Dipti[1], Mohammad Razuanul Hoque[4], Hiroaki Taniguchi[5], M. Rezaul Karim[1]***

1 Laboratory for Cancer Biology, Department of Biotechnology and Genetic Engineering, Jahangirnagar University, Savar, Dhaka, Bangladesh, 2 Department of Hepatology, Bangabandhu Sheikh Mujib Medical University, Dhaka, Bangladesh, 3 Biomedical Research Foundation, Dhaka, Bangladesh, 4 Department of Biochemistry and Molecular Biology, University of Chittagong, Chittagong, Bangladesh, 5 Institute of Genetics and Animal Breeding of the Polish Academy of Sciences, Jastrzebiec, Magdalenka, Poland

☯ These authors contributed equally to this work.
* mrkarimcu@gmail.com

**Data Availability Statement:** All relevant data are within the manuscript and its Supporting Information files.

## Abstract

Hepatocellular carcinoma (HCC) is the sixth most common cancer and the third most common cause of cancer mortality worldwide. Infection with hepatitis B virus (HBV) and/or hepatitis C virus (HCV) is the most predominant cause of HCC. Concerns arise for the presence of additional risk factors, as there is still a large proportion of patients without HBV or HCV infection. Previous studies have reported that higher intake of fruits and vegetables and reduced consumption of red/processed meat might play a protective role in HCC etiology, though the nationwide proof is limited. Hence, we studied multiple risk factors including food habit, lifestyle, and clinical implications of HCC patients in Bangladeshi. Demographic, clinical, and biochemical data, as well as data on food habits, were collected in this study. Our results indicated that a high intake of rice (AOR 4.28, 95% CI 1.48 to 14.07, p = 0.011), low intake of fruits (AOR = 4.41 95% CI 1.48–15.46; p = 0.012), leafy vegetables (AOR = 2.80, 95% CI 1.32–6.08; p = 0.008), and fish (AOR = 4.64 95% CI 2.18–10.23; p<0.001) increased the HCC risk. Moreover, a high intake of eggs (AOR = 2.07 95% CI 0.98–4.43; p = 0.058) also showed an increased risk. Roti, non-leafy vegetables, red meat, and tea were found to have no association with HCC risk. This study revealed that food habit patterns and lifestyle may have a profound effect on HCC development among Bangladeshi patients in addition to well established risk factors.

## Introduction

Cancer that first develops in the liver is called primary liver cancer, and hepatocellular carcinoma (HCC) is the most common one among them. 75%- 85% of primary liver cancers are

**Funding:** M. Rezaul Karim Grant no. 4308 University Grant Commission (UGC) of Bangladesh http://www.ugc.gov.bd/, and Dr. Hiroaki Taniguchi Grant no. OPUS 13 (2017/25/B/NZ5/02762) The Polish National Science Center. The funders had no role in study design, data collection and analysis, decision to publish, or preparation of the manuscript.

**Competing interests:** The authors have declared that no competing interests exist.

attributed to HCC, while 10%-15% account for intrahepatic cholangiocarcinoma (ICC), and the residual cases include other rare types [1]. The international agency for research on cancer estimated approximately 0.84 million incidences and 0.78 million mortalities worldwide in 2018, which will rise to 1.36 million incidences and 1.28 million deaths by the year 2040. Liver cancer is the sixth most commonly diagnosed cancer, and it was the third cause of mortality worldwide in 2018 among all sexes. Rates of both incidence and mortality are 2 to 3 times higher among men in most of regions worldwide [2]. Dynamic temporal trends, marked variations among geographic regions, racial and ethnic groups, and between men and women, and the presence of several well-documented environmental potentially preventable risk factors are several exciting epidemiologic features of HCC [3].

Several studies have shown that HCC occurs worldwide, mostly due to infection with hepatitis B virus (HBV). The other recognized risk factors for HCC include chronic hepatitis C virus (HCV) infection, exposure to dietary aflatoxin, smoking, fatty liver disease, excessive alcohol consumption, diabetes, etc. [4]. Recent evidence found a positive correlation between obesity and increased risk of liver cancer through chronic inflammation [5]. Another risk factor is diabetes, which may also increase the risk of liver cancer [6]. The major risk factors vary from region to region [1]. In China and Africa, which are the most high-risk HCC areas, the major risk factors are chronic HBV infection and aflatoxin. On the other hand, in Japan and Egypt, the primary risk factor is attributed to HCV infection [7,8]. People from low-risk HCC areas are also prone to obesity and diabetes as a risk factor [9].

Bangladesh is one of the countries facing a considerable burden of HCC. HCC is the third most common cancer in the country, and it is just behind lung cancer and stomach cancer [10,11]. HBV infection is the leading cause of HCC in Bangladesh, which is estimated at 46.9% to 61% [13,14]. However, there is not much information on the definite risk factors of HCC and no appropriate and reliable data showing the etiological and epidemiological perspectives yet.

Furthermore, there is no available data regarding the Bangladeshi population to understand the cause of developing HCC in those patients who do not have any HBV or HCV infection. Apart from the most common and defined etiological factors, the association of diet with the development of HCC is unclear [10]. Previously, few studies were investigating the role of food habits in the development of HCC worldwide. It is evident from those studies that food habits may play an important role in HCC development. Different studies claimed an inverse relation of milk [11], fiber and whole-grains [12], white meat [13], fruits and vegetables [14], and fish [15] with HCC. In contrast, a high intake of red meat [13] has been associated with increased risk of HCC; however, there is no sufficient evidence on the association between HCC and egg consumption [16]. Therefore, intense investigations are required to develop insights about clinical features, etiological factors, and epidemiology of HCC in Bangladesh. In this study, we took an elaborate history of HCC patients, details of food habit patterns, smoking and drinking habits, and also checked clinical profiles and etiological agents of HCC patients. Our study sheds light on the association of HBV and HCV infection for the development of HCC, as well as some other risk factors like food habit patterns, and smoking and drinking habits for the development of HCC in Bangladeshi patients.

## Materials and methods

### Study population

In the present hospital-based case-control study, we observed 80 patients with HCC who attended a tertiary care facility from November 2018 to July 2019. This facility is the largest postgraduate public medical university in Bangladesh, located in the capital city, Dhaka, and

acts as a reference center for patients having unmanageable diseases. After matching the demographic characteristics of HCC patients, such as age, sex, income, and sociodemographic status, 101 control subjects were chosen. Among the patients visiting public hospitals in Bangladesh, most are the poor and lower-middle-income groups. Hence, we assumed that only this group of people represented the study population in this particular research study. This study was approved by the ethical committee of that particular tertiary hospital and was conducted according to the European Association for the Study of the Liver (EASL) clinical practice guidelines for the management of hepatocellular carcinoma [17]. We clarified to the participants the purpose and procedure of the study in detail, their benefits and risks, and subsequently informed consents were obtained from both patients and controls.

## Study design and sample size calculation

A matched case-control design was used for this study. The sample size was calculated based on a conservative predictor such as intake of leafy vegetables and whether a low amount of intake compared to moderate intake increases the odds of HCC. Assuming a prevalence of the risk factor in the unexposed population to be 50%, to detect an odds ratio (OR) of at least 2.5 with 80% power with 95% confidence would require 162 subjects. The sample size was calculated using the R package epiR [18].

## Patient selection

The inclusion criteria for patient choice include both male and female patients with HCC regardless of etiology. The exclusion criteria included a) patients with additional cancer as well as HCC and b) patients with co-morbid conditions such as severe congestive cardiac failure (CCF), ischemic heart disease (IHD), chronic kidney disease (CKD), etc., and not fit for fine-needle aspiration cytology (FNAC).

## Diagnosis procedure

HCC has been diagnosed on the grounds of clinical and radiological characteristics (ultrasonography and computed tomography), followed by EASL clinical practice guidelines [17]. The confirmation of HCC was done by cytopathology examination, collecting tissues through fine-needle aspiration cytology (FNAC) technique [19,20]. Patients under 18 years of age were excluded.

## Clinical and biochemical evaluation

All patients were clinically assessed, and blood pressure level and Body Mass Index (BMI) were recorded. Patients having a BMI of $>25$ kg/m$^2$ were marked as obese, and patients with a BMI of $<25$ kg/m$^2$ were considered as non-obese. Patients' blood samples were drawn under fasting conditions, and the accompanying tests, for example, complete blood count (CBC), albumin, total bilirubin, alanine aminotransferase (ALT), aspartate aminotransferase (AST), international normalized ratio (INR), and alpha-fetoprotein (AFP) were performed for diagnosis purpose. Barcelona Clinic Liver Cancer (BCLC) staging and Child-Pugh scores were determined from laboratory tests, and clinical features obtained from diagnostic reports. Patients were categorized into four BCLC stages, for example, Early-stage, A; Intermediate stage, B; Advanced stage, C; and Terminal stage, D. BCLC staging was determined by physician, and patients' performance status, tumor size and number, Child-Pugh score and portal vein involvement [17,21] were documented according to EASL guidelines.

## Data collection

We collected patients' demographic, clinical, and biochemical information through an interview with a structured questionnaire. Demographic information, such as age, sex, education, earnings, food habits, HCC etiologies, and first presenting symptoms were collected. Diverse clinical and biochemical information such as serum levels of total bilirubin, albumin, INR, AFP, aspartate aminotransferase, alanine aminotransferase, presence of ascites, hepatic encephalopathy, hepatomegaly, splenomegaly, was collected from patient's diagnostics reports. A liver radiologist deliberately looked into patients' computed tomography (CT), and ultrasonography report and the size, area, number of tumor lesions, portal vein thrombosis were noted. Child-Pugh classification and BCLC staging of the patients were recorded.

## Assessment of dietary habit

The data on food consumption per capita were obtained by an interview-based, structured questionnaire. The survey incorporated the food habit pattern from both cases (n = 80) and controls (n = 101). Through a case-control statistical analysis, we explored the link between food habit patterns and HCC development. In the case of fruits and vegetables, seasonal consumption and the corresponding duration are subject to variation. The dietary items included 82 foods or food groups and were divided into 9 sections: i) rice (primary course); ii) bread, and roti (secondary course); iii) leafy vegetables (water spinach, pumpkin leaves, taro stem, Indian spinach, spinach, red amaranth, cauliflower, cabbage); iv) non-leafy vegetables (okra, tomato, balsam apple, eggplant, carrot, pumpkin, potatoes, sweet potatoes); v) meat and meat-based food items such as burger, sandwich; vi) fish (both river and ocean); vii) milk, tea, coffee, sugar, tea with condensed milk; viii) fruit (litchis, mangoes, jackfruits, blackberries, dates, guavas, pineapple, papayas, bananas, watermelon, coconuts, apples, grapes, oranges, tropical fruits, etc.); ix) sweets, rice-based desserts, and soft drinks. The selection of food items was based on foods regularly consumed by the Bangladeshi people. The standard serving size was obtained from the dietary guidelines from BIRDEM (Bangladesh Institute of Research and Rehabilitation in Diabetes, Endocrine, and Metabolic Disorders) [22]. We focused on the above-mentioned food items to explore the relationship between these food groups and the advancement of HCC in the Bangladeshi population. The consumption rate among our case-control population of targeted food groups was transformed into g/day or ml/day. In the case of tea it was considered as cup/day. Consumption of specific food items of more than the suggested value is defined as "high intake."

In contrast, the consumption of specific food items of lower than the suggested value is defined as "low intake." A recent report suggested an inverse association of tea intake with primary liver cancer; however, the preparation of tea in Bangladesh or South Asia is different compared to Western countries. In Bangladesh, condensed milk with sugar is commonly used to prepare the tea. Moreover, the tea leaves are usually continuously boiled for a long time. Hence, we investigated the primary or combined effect of tea drinking on the risk of HCC with or without the presence of other risk factors.

## Statistical analysis

Data management and statistical analysis were performed using R statistical software [18]. Continuous variables were expressed as mean ± standard deviation, and categorical variables were presented as numbers and percentages or frequencies. The Chi-square ($\chi2$) test with continuity correction was employed to find significant differences between groups. Crude and adjusted odds ratios were calculated using the multiple logistic regression model. For the regression strategy, we first performed bivariate analysis of the potential factors with the

outcome variable of interest. If the bivariate results were significant at 20% level, we considered them in the regression model. In addition, certain variables were included in the model for their importance from demographic and clinical perspectives regardless of the results of the bivariate analysis. These included age, sex, diabetes status, and weight status (overweight or normal). The analysis was performed to assess the effect of risk factors on the likelihood of developing HCC. A $p$-value <0.05 was considered as statistically significant.

## Results

### Demographic characteristics of the study sample

In the present study, patients of both sexes, with different age groups, social, marital, educational, and professional backgrounds have been selected (Table 1). Our sample consisted of 80 study patients and 101 healthy controls. Of the study subjects, 64 were male, and 16 were female with a calculated sex ratio of 4:1. The mean age of the study population was 48.7±14.8 years and 47.1±14.2 years for the controls. The mean Body Mass Index (BMI) was calculated as 23.8±2.6 for the study group and 23.7±2.7 for the controls. For both study and control groups, there were more patients with no education, followed by patients with the primary education level completed. These two categories comprised two- thirds of the total study population. A large subset of patients (20 cases and 23 controls) was employed by the government or in non-government services. Regarding the employment, the highest category (21 patients) was found for people involved in different kinds of businesses, and 15 patients were farmers. Among 16 women in the study population, 14 were housewives. While we shed light on economic solvency based on income level, roughly 58% of the patients had an income between 20,000–50,000 BDT per month. More than 90% of the patients were married, and about 73% lived in rural areas.

### Etiological factors

Fig 1 outlines the etiological variables of our study sample. As it is already recognized that the most common etiology for HCC is cirrhosis of the liver, 78% (62) of our HCC patients also reported cirrhosis leading to HCC. Next, we found smoking (50% reporting) as a risk factor followed by HBV infection, which accounted for 39 (49%) of our patients. We found only 6 (8%) patients having HCV infection. There were no patients diagnosed jointly infected with both hepatitis B and hepatitis C viruses. Surprisingly, 35 (44%) patients were found to be negative for both hepatitis B and hepatitis C infections. 23 (29%) of the patients were diabetic while 17 (21%) had both diabetes and positive HBV/HCV infection. While smoking together with hepatitis B was the sole etiology for 22 (28%) patients, only 3 (4%) patients had the etiological history of smoking along with hepatitis C. Consumption of alcohol alone as etiology accounted for only 3 patients (4%). Alcohol consumption, however, is not very common in Bangladesh due to religious restriction and strict state law. Only 2 (2%) patients with alcohol intake along with hepatitis B were registered. However, there were no patients with hepatitis C reporting alcohol consumption. Given a large proportion of cirrhosis positive HCC patients with smoking behavior, we compared the smoking behavior and development of cirrhosis (Table 2). The result suggests the odds of developing cirrhosis for those who smoke is more than four times compared to those who do not smoke.

### First clinical symptoms of HCC patients

The first clinical symptoms of HCC patients are shown in Fig 2. Practically the majority of the patients in our study were symptomatic. In most cases, one patient had multiple symptoms.

**Table 1. Sociodemographic and food intake characteristics of HCC patients and the controls.**

| Label | Levels | Control | HCC | p |
|---|---|---|---|---|
| **Total N (%)** | | 101 (55.8) | 80 (44.2) | |
| **Age in years** | Mean (SD) | 47.1 (14.2) | 48.7 (14.8) | **0.461** |
| **Sex** | F | 20 (19.8) | 16 (20.0) | **0.974** |
| | M | 81 (80.2) | 64 (80.0) | |
| **BMI** | Mean (SD) | 23.7 (2.7) | 23.8 (2.6) | **0.988** |
| **Education (level completed)** | No Education | 41 (40.6) | 36 (45.0) | **0.748** |
| | Primary | 29 (28.7) | 25 (31.2) | |
| | Secondary | 9 (8.9) | 8 (10.0) | |
| | Higher Secondary | 12 (11.9) | 6 (7.5) | |
| | Graduate | 10 (9.9) | 5 (6.2) | |
| **Occupation** | Business | 34 (33.7) | 21 (26.2) | **0.883** |
| | Farmer | 17 (16.8) | 15 (18.8) | |
| | Housewife | 16 (15.8) | 14 (17.5) | |
| | Labor and Other | 11 (10.9) | 10 (12.5) | |
| | Service | 23 (22.8) | 20 (25.0) | |
| **Income (BD Taka)** | 1: 1–20,000 | 7 (6.9) | 4 (5.0) | **0.784** |
| | 2: 20,001–50,000 | 60 (59.4) | 46 (57.5) | |
| | 3: > 50,000 | 34 (33.7) | 30 (37.5) | |
| **Place of residence** | Rural | 75 (74.3) | 58 (72.5) | **0.790** |
| | Urban | 26 (25.7) | 22 (27.5) | |
| **Marital status** | Married | 91 (90.1) | 77 (96.2) | **0.111** |
| | Unmarried | 10 (9.9) | 3 (3.8) | |
| **Smoker** | No | 71 (70.3) | 40 (50.0) | **0.005** |
| | Yes | 30 (29.7) | 40 (50.0) | |
| **Diabetes** | No | 75 (74.3) | 57 (71.2) | **0.651** |
| | Yes | 26 (25.7) | 23 (28.7) | |
| **Rice intake** | High | 72 (71.3) | 74 (92.5) | **<0.001** |
| | Moderate | 29 (28.7) | 6 (7.5) | |
| **Roti/Bread** | Low | 66 (65.3) | 61 (76.2) | **0.111** |
| | Moderate | 35 (34.7) | 19 (23.8) | |
| **Egg intake** | High | 38 (37.6) | 50 (62.5) | **0.001** |
| | Moderate | 63 (62.4) | 30 (37.5) | |
| **Tea with condensed milk** | No | 49 (48.5) | 33 (41.2) | **0.330** |
| | Yes | 52 (51.5) | 47 (58.8) | |
| **Tea with tea-bag** | No | 76 (75.2) | 62 (77.5) | **0.724** |
| | Yes | 25 (24.8) | 18 (22.5) | |
| **Leafy vegetable intake** | Low | 43 (42.6) | 55 (68.8) | **<0.001** |
| | Moderate | 58 (57.4) | 25 (31.2) | |
| **Non-leafy vegetable intake** | Low | 48 (47.5) | 36 (45.0) | **0.735** |
| | Moderate | 53 (52.5) | 44 (55.0) | |
| **Fruit intake** | Low | 73 (72.3) | 75 (93.8) | **<0.001** |
| | Moderate | 28 (27.7) | 5 (6.2) | |
| **Fish intake** | Low | 38 (37.6) | 62 (77.5) | **<0.001** |
| | Moderate | 63 (62.4) | 18 (22.5) | |
| **Milk intake** | Low | 55 (54.5) | 66 (82.5) | **<0.001** |
| | Moderate | 46 (45.5) | 14 (17.5) | |
| **Red meat intake** | High | 10 (9.9) | 6 (7.5) | **0.572** |

(*Continued*)

**Table 1.** (Continued)

| Label | Levels | Control | HCC | p |
|---|---|---:|---:|---:|
| | Moderate | 91 (90.1) | 74 (92.5) | |
| **White meat intake** | Low | 35 (34.7) | 40 (50.0) | **0.037** |
| | **Moderate** | **66 (65.3)** | **40 (50.0)** | |

The most significant number of patients were recorded with upper abdominal pain, nearly 70 (88%). Around 67 (84%) patients lost weight during the disease period. A large proportion of HCC patients had anorexia, about 62 (78%). There were some other complications such as pain in the right shoulder, fever, headache, and vomiting reported by 42 (52%), 51 (64%), 36 (45%), and 33 (41%) patients, respectively.

## Clinical characteristics of HCC patients

The medical records of 73 patients were reviewed and illustrated in Table 3. Many patients were unable to bear the costs of all biochemical and radiological diagnosis due to poor financial conditions. Some patients were not willing to provide their medical records. As a consequence, every patients' diagnosis report was not accessible. Therefore, we mention the actual numbers of each parameter, which is the total number of HCC. In our cohort of study, the majority of patients 43 (66.2%) had a single nodule. Many of them 26 (45.6%) had a nodule size of 5–10 cm. Different clinical features were depicted by patients, with 50 patients with hepatomegaly (72.5%), followed by 31 patients (44.9%) with splenomegaly. In addition, mild ascites was observed in 27 patients (38.6%), and 39 patients (55.7%) showed no ascites. Portal vein thrombosis was noted distinctly for 17 patients (24.6%) and 14 patients (19.4%) showed mild encephalopathy. The Child-Pugh score is a system determined by scoring five clinical

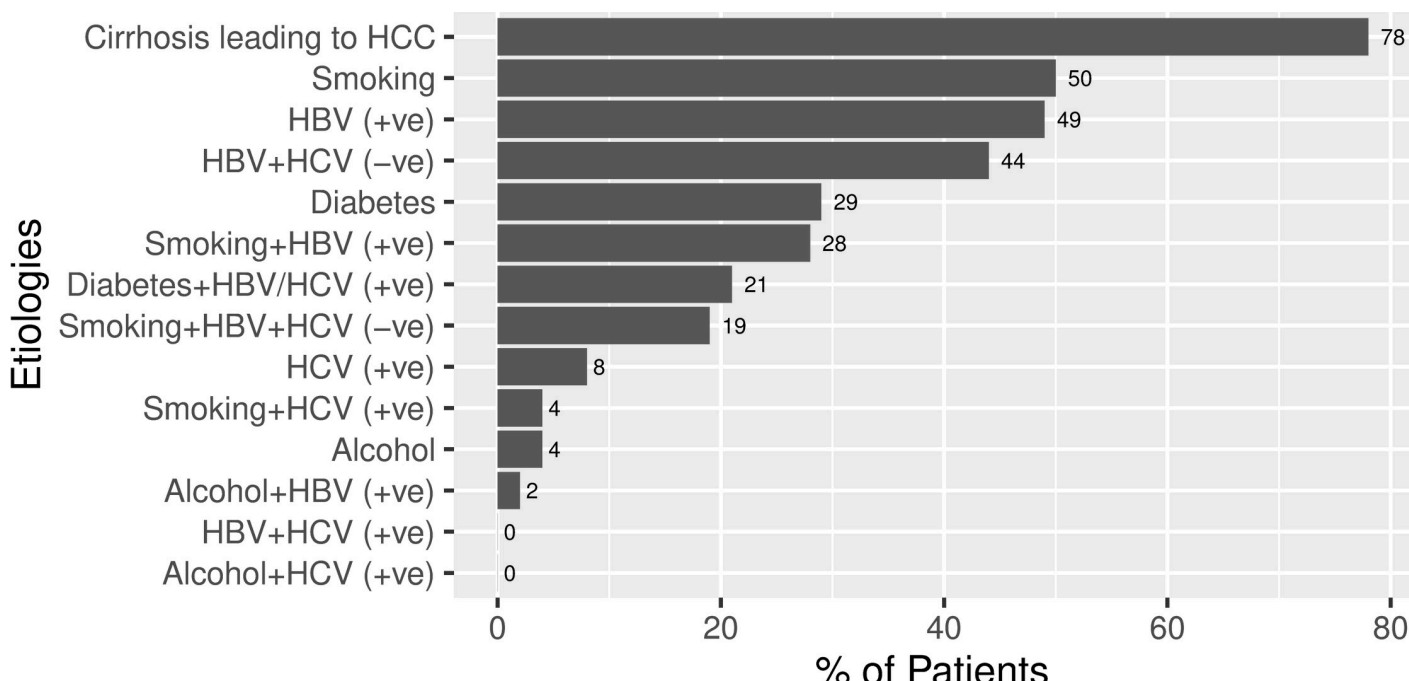

**Fig 1. Underlying etiologies of Bangladeshi HCC patients.** HBV = hepatitis B virus, HCV = hepatitis C virus.

Table 2. Smoking behavior between cirrhosis positive and cirrhosis negative HCC patients.

|  | Cirrhosis | | |
|  | No | Yes | Total |
| --- | --- | --- | --- |
| No | 14 (77.8) | 26 (41.9) | 40 (50.0) |
| Yes | 4 (22.2) | 36 (58.1) | 40 (50.0) |
| Total | 18 | 62 | 80 |

measures such as total bilirubin, serum albumin, prothrombin time, ascites, and hepatic encephalopathy for evaluating the prognosis of liver disease. It serves as a liver function marker and helps to determine appropriate treatment. Child-Pugh class A, B, and C are classified based on severity, with C being the most severe. As indicated by our study, most patients had Child-Pugh class A, 35 (47.9%), and B, 31 (42.5%).

## Barcelona Clinic Liver Cancer (BCLC) staging and treatment

The magnitude of the tumor has been staged according to the BCLC classification (Table 4). Early HCC (stage A) was categorized in 4 patients (5.5%). Furthermore, 40 patients (54.8%) and 27 patients (37%) were categorized respectively as stages B and C. A total of 2 patients (2.7%) were categorized as stage D. Table 3 shows the treatment modalities used in our patients at various BCLC stages. Patients underwent ablation in stage A, and chemoemboliza-tion in stage B. On the other hand, patients in stage C underwent systemic therapy and clini-cians suggested best supportive care (BSC) in the event of stage D.

## Alpha-fetoprotein (AFP) levels as a marker for HCC

AFP is a tumor marker protein that can increase significantly in case of liver damage and certain other cancers. In our study we have documented the AFP levels of 73 patients, which have been shown in Table 5. As indicated by our study, 4 (100%) patients with BCLC stage A had AFP levels greater than 1000 ng/ml. In the case of stages B and C, AFP levels were found to be higher than 1000 ng/ml for 20 (50%) and 18 (66.6%) patients, respectively, whereas for 19 (47.5%) and 7 (26.0%) patients, levels lower than 200 ng/ml were observed. Two out of 73 patients had BCLC stage D with AFP levels greater than 1000 ng/ml.

## Association of food habits with the risk of HCC

In the present study, we intended to focus on the association between food habits and the risk of HCC in Bangladesh. An approved food frequency questionnaire (FFQ) was used to evaluate the habitual diet of the study subjects and healthy controls. To select the variables for the regression model, we used predictors that were found to be significant at the alpha = 10% level in the bivariate analysis. However, to further control demographic confounders, we kept age and sex in the model, although they are not significantly different between the two groups.

Table 6 represents the ordered distribution of the total number of individuals consuming low, moderate, or high amounts of particular food items, and the crude and adjusted odds ratios with 95% confidence intervals for HCC. Adjusted for other factors, the odds of developing HCC was 4.34 times for those with high rice intake compared to moderate intake (95% CI: 1.49–14.42, p = 0.010). Similarly, low intake of leafy vegetables (AOR 2.8, 95% CI 1.3 to 6.03, p = 0.009), low fruit intake compared to moderate (AOR 4.40, 95% CI: 1.47–15.51, p = 0.012), and low fish intake compared to moderate (AOR 4.64, 95% CI: 2.18–10.26, p<0.001) increase the odds of HCC significantly. On the other hand, no significant association was found in the case of white meat, milk intake, having diabetes, and weight status (obese or normal). Age and

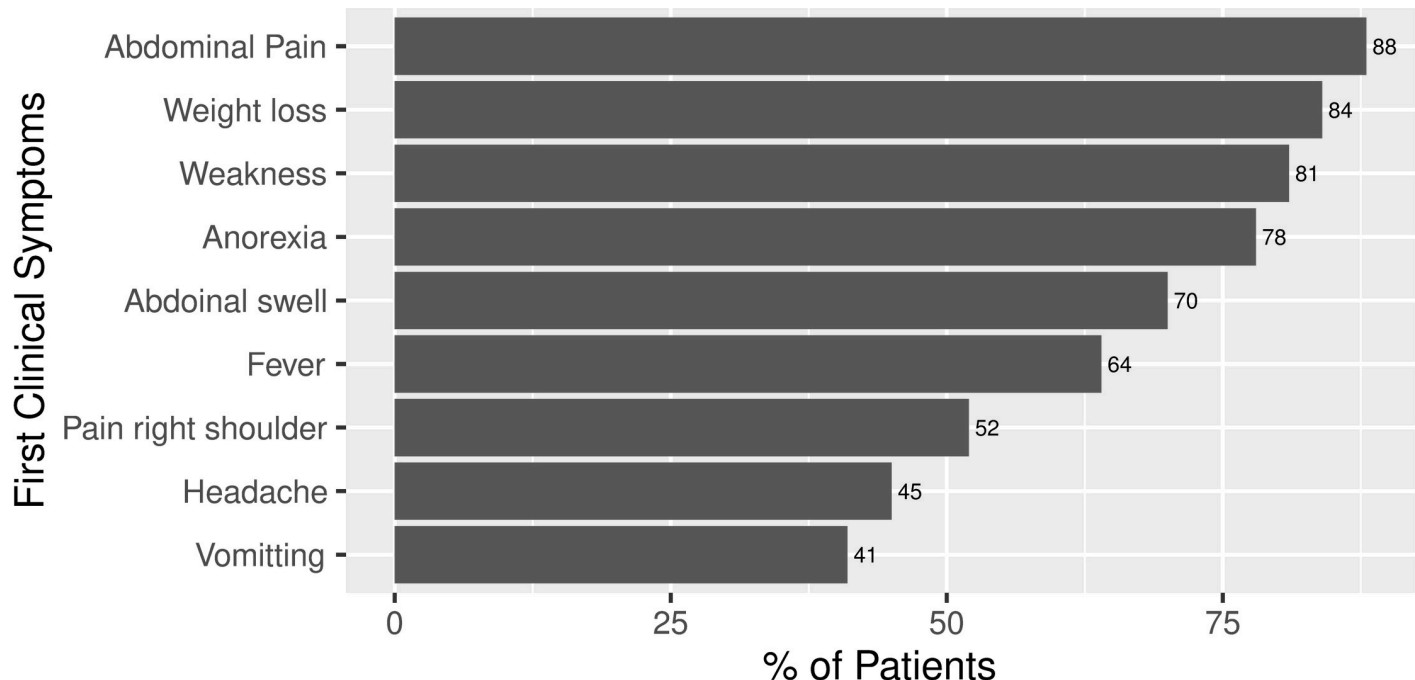

**Fig 2. Percentage distribution of first clinical symptoms of HCC patients.** Multiple responses possible, therefore the percentages will not add up to 100%.

**Table 3. Clinical characteristics of HCC patients.**

| Parameter | | Patients, N (%) |
|---|---|---|
| Tumor size (cm) | Mean (SD) | 8.4 (3.7) |
| Tumor size (binned) | <5 cm | 13 (22.8) |
| | 5–10 cm | 26 (45.6) |
| | >10 cm | 18 (31.6) |
| Number of tumors | Single | 43 (66.2) |
| | Multiple | 22 (33.8) |
| Portal vein thrombosis | No | 52 (75.4) |
| | Yes | 17 (24.6) |
| Hepatomegaly | No | 19 (27.5) |
| | Yes | 50 (72.5) |
| Splenomegaly | No | 38 (55.1) |
| | Yes | 31 (44.9) |
| Ascites | None | 39 (55.7) |
| | Mild | 27 (38.6) |
| | Severe | 4 (5.7) |
| Encephalopathy | None | 58 (80.6) |
| | Mild | 14 (19.4) |
| | Severe | 0 (0.0) |
| Child-Pugh score | A | 35 (47.9) |
| | B | 31 (42.5) |
| | C | 7 (9.6) |

**Table 4. Barcelona Clinic Liver Cancer staging (BCLC) and treatment.**

| Label | Levels | Ablation | BSC | Chemoembolization | Systemic therapy | Total |
|---|---|---|---|---|---|---|
| BCLC stage | A | 4 (100.0) | 0 (0.0) | 0 (0.0) | 0 (0.0) | 4 (5.5) |
| | B | 0 (0.0) | 0 (0.0) | 40 (100.0) | 0 (0.0) | 40 (54.8) |
| | C | 0 (0.0) | 0 (0.0) | 0 (0.0) | 27 (100.0) | 27 (37.0) |
| | D | 0 (0.0) | 2 (100.0) | 0 (0.0) | 0 (0.0) | 2 (2.7) |

sex were matched for the study, and as a result, they demonstrated no significance on the risk of HCC.

Interestingly, this study found 35 (44%) patients negative for both hepatitis B and hepatitis C infections. Hence, it is important to investigate whether any particular food group is primarily responsible for potential HCC risk. However, we did not observe any statistically significant difference in food habits, when comparing the patients who were hepatitis B or C positive with the rest of the patients (categorized as 'Other') (S1 Table).

## Discussion

In the present study, we characterized HCC patients in Bangladesh according to HBV and HCV infection, as well as some other risk factors like food habit patterns, smoking, and drinking habits for the first time regarding the Bangladeshi perspective. We represented unique profiles in terms of demographic factors, etiologies, disease-specific clinical representation, staging and treatment options, biomarker profiles, and an association of food habit patterns with the development of HCC for those patients. This represents the first-ever elaborated study accompanying factors other than HBV and HCV infection as a predominant cause of HCC in Bangladesh.

Among the 80 patients we studied, the male predominance was found, which is not surprising as it is evident that males are more susceptible to HCC than females [23]. In almost all populations, male to female ratios usually average between 2:1 and 4:1 [3]. In our study, the male to female ratio was found to be 4:1. This is due to the sex-specific differences in exposure to the risk factors, because they are more likely to be infected with HBV and HCV, as well as alcohol consumption, cigarette smoking, and food habits. The liver is a hormone-sensitive organ, that's why sex hormones, such as androgen and estrogen, maybe an acting factor. It is assumed that androgen promotes HCC development, whereas estrogen plays a protective role [24–26]. The typical age group affected by HCC was 50–59 and 60–69, as we have found most patients corresponded to this age group. In a study performed in Bangladesh, a group found 41 to 50 years as the most common age group to develop HCC [27]. In the United States, from the year 1992 to 2013 the age-specific incidence rate was highest in the age group of 50–69. However, a significant number of patients were also found above the age of 70 [28]. The mean age of our study population was 48.71±14.8 years. The study conducted by Gani et al. (2013) comprised

**Table 5. The range of AFP levels according to BCLC staging.**

| Label | Levels | <200 | 200–1000 | >1000 | Total |
|---|---|---|---|---|---|
| BCLC stage | A | 0 (0.0) | 0 (0.0) | 4 (100) | 4 (5.5) |
| | B | 19 (47.5) | 1 (2.5) | 20 (50) | 40 (54.8) |
| | C | 7 (26.0) | 2 (7.4) | 18 (66.6) | 27 (37.0) |
| | D | 0 (0.0) | 0 (0.0) | 2 (100) | 2 (2.7) |

Normal range of AFP < 15ng/ml

**Table 6. Risk factors associated with HCC: Results of multiple linear logistic regression analysis.**

| Dependent: Subject | | Control | HCC | OR (univariable) | OR (multivariable) |
|---|---|---|---|---|---|
| Age | Mean (SD) | 47.1 (14.2) | 48.7 (14.8) | 1.01 (0.99–1.03, p = 0.455) | 1.01 (0.98–1.04, p = 0.445) |
| Sex | F | 20 (55.6) | 16 (44.4) | - | - |
| | M | 81 (55.9) | 64 (44.1) | 0.99 (0.47–2.08, p = 0.974) | 0.87 (0.33–2.35, p = 0.783) |
| Smoker | No | 71 (64.0) | 40 (36.0) | - | - |
| | Yes | 30 (42.9) | 40 (57.1) | 2.37 (1.29–4.40, p = 0.006) | 1.73 (0.78–3.86, p = 0.180) |
| Rice intake | Moderate | 29 (82.9) | 6 (17.1) | - | - |
| | High | 72 (49.3) | 74 (50.7) | 4.97 (2.07–13.90, p = 0.001) | 4.34 (1.49–14.42, p = 0.010) |
| Egg intake | Moderate | 63 (67.7) | 30 (32.3) | - | - |
| | High | 38 (43.2) | 50 (56.8) | 2.76 (1.52–5.11, p = 0.001) | 2.08 (0.98–4.48, p = 0.059) |
| Leafy vegetable intake | Moderate | 58 (69.9) | 25 (30.1) | - | - |
| | Low | 43 (43.9) | 55 (56.1) | 2.97 (1.62–5.56, p = 0.001) | 2.76 (1.30–6.03, p = 0.009) |
| Fruit intake | Moderate | 28 (84.8) | 5 (15.2) | - | - |
| | Low | 73 (49.3) | 75 (50.7) | 5.75 (2.28–17.66, p = 0.001) | 4.40 (1.47–15.51, p = 0.012) |
| Fish intake | Moderate | 63 (77.8) | 18 (22.2) | - | - |
| | Low | 38 (38.0) | 62 (62.0) | 5.71 (3.00–11.30, p<0.001) | 4.64 (2.18–10.26, p<0.001) |
| Milk intake | Moderate | 46 (76.7) | 14 (23.3) | - | - |
| | Low | 55 (45.5) | 66 (54.5) | 3.94 (2.00–8.14, p<0.001) | 2.05 (0.87–4.95, p = 0.104) |
| White meat intake | Moderate | 66 (62.3) | 40 (37.7) | - | - |
| | Low | 35 (46.7) | 40 (53.3) | 1.89 (1.04–3.45, p = 0.038) | 1.64 (0.75–3.59, p = 0.214) |
| Diabetes | No | 75 (56.8) | 57 (43.2) | - | - |
| | Yes | 26 (53.1) | 23 (46.9) | 1.16 (0.60–2.25, p = 0.651) | 1.01 (0.42–2.37, p = 0.991) |
| Weight status | Normal | 73 (53.7) | 63 (46.3) | - | - |
| | Overweight | 28 (62.2) | 17 (37.8) | 0.70 (0.35–1.39, p = 0.318) | 0.81 (0.33–1.98, p = 0.649) |

Number in data set = 181, Number in model = 181, Missing = 0, AIC = 196.9, C-statistic = 0.857, H&L = Chi-sq(8) 6.38 (p = 0.605)

57 HCC patients where the mean age was 45.81 ± 15.31 years. The educational background represents the lack of awareness and knowledge, as a large proportion of patients had no education. According to the income level, most of the patients were of lower or lower-middle-income group and most of the patients were of rural origin. These represent the economic constraints to accessing necessary tests and treatments as well as the lack of suitable medical care in rural areas. HCC has always been a very much neglected disease in Bangladesh as most of the patients lack the economic capability and they also do not have proper knowledge about HCC. Other risk factors are a lack of medical facilities and delay of diagnosis [29].

Chronic HBV and HCV infection is considered as one of the leading causes of the appearance of HCC in Bangladesh [29,30]. Despite the introduction of vaccination during 2003–2005 into the Expanded Program on Immunization (EPI) in Bangladesh, HBV infection remains abundant in the middle and older age adult population. Our data showed that chronic HBV was the significant risk factor contributing to the development of HCC, which is similar to the previous report [29] and also similar with our neighboring country India [31]. Although HCV is considered one of the leading causes of the appearance of liver cancer in many countries [32,33], in our study, HCV infection was found in only 8% of patients. Alcohol consumption is another risk factor of HCC in Western countries [34,35]. In our study, only 4% of patients, including one patient with HBV infection, were found consuming alcohol as alcohol consumption is strictly restricted in Bangladesh by state law. Alcohol consumption is also very much restricted due to socioeconomic conditions and religious restrictions. According to our study, we found no significant value for alcohol as an individual risk factor. Donato et al. [36]

examined the association between alcohol intake and HCC and found that for each level of alcohol intake, the highest risks were observed among subjects with HCV infection, followed by those with HBV infection, and finally by those without hepatitis virus infection. So, alcohol consumption is not an independent important risk factor in Bangladesh. In our study, we found 50% of patients were chain smokers and most of them had no HBV and HCV infection. The relationship between cigarette smoking and HCC has been examined in many studies. In almost all studies, both positive-association [37–39] and lack-of-association [38] have been reported.

According to our study, smoking persisted as an independent risk factor in our country. The effect of smoking was also found to be an independent risk factor for HCC in previous studies [40,41]. Furthermore, smoking has been reported to increase the risk of development of HCC in people with HCV, HBV [42]. A synergistic combined effect of HBV/HCV and smoking might act mostly through increased risk of HCC (Fig 1). So, there could be a synergistic interaction between tobacco smoking and HBV/HCV for the development of HCC in Bangladeshi patients. In our study we found 29% of patients were diabetic. The association between diabetes mellitus type II was found as an individual risk factor in other studies [43]. Though diabetes did not emerge as a strong individual risk factor in our study, we found a synergistic combined effect of HBV/HCV and diabetes for the development of HCC (Fig 1). Nevertheless, we did not observe any impact of obesity on the odds of HCC in our multivariate logistic regression model (Table 6).

In our study, we collected the first clinical symptoms that have been observed while the patients were admitted to the hospitals. It was evident that every patient came with multiple clinical symptoms which were representative symptoms of HCC. The habit of Bangladeshi patients avoiding clinical checkups and regular screening of disease has become a significant influence on the development of devastating diseases, including HCC. The major factors behind this are poverty and lack of education and awareness, which we have already mentioned. After being admitted to hospital, different imaging reports such as CT scan, ultrasonographies, biochemical tests, and tumor markers were used for the detection of presence or severity of HCC. In Bangladesh, as mentioned before the diagnosis of the disease is usually delayed. As a result, the patients developed in BCLC stage B (intermediate stage) and C (advanced stage). Most of the patients of our study were in stage B and stage C. The BCLC staging was determined according to the patients' albumin, bilirubin, ascites, encephalopathy, imaging reports, tumor size, portal vein invasion, tumor site, Child-Pugh score, and prothrombin activity.

The BCLC staging helps in the decision of treatment options available for the patients by the clinicians. It narrows down the treatment options to provide particular treatments to the patients. Due to resource constraints and lack of treatment facilities, in addition to the limitation mentioned above, the treatment options were optimized to the most affordable options. Commonly, the patients of stage A were given ablation therapy, stage B patients were given chemoembolization, and stage C patients were given chemotherapy. Liver transplantation is not a popular treatment option in Bangladesh due to its high cost and other difficulties. Clinicians also try to avoid providing these treatment options to the patients. If any patients needed surgery, they were transferred to the surgery department. But in most of the cases, the hepatology department provides the ablation, chemoembolization, and chemotherapy without referring the patients to the surgery. In addition to the BCLC staging, AFP level which is a tumor marker helps in the diagnosis of HCC as well as provide insights into the treatment pattern.

Although the role of diet in the etiology of hepatocellular carcinoma is unclear, evidence suggests that dietary intake of particular food groups may have a favorable or adverse association with the risk of HCC [44]. For example, consumption of vegetables and fruits may have

an inverse connection, while consumption of red meat can increase the risk of HCC. Not many studies have been evaluated to explore the link between food intake and liver cancer, either in our country or globally. In our study, analysis of dietary habits of HCC patients and controls indicated that diet has a relevant role in HCC risk. Rice is the staple food of the Bangladeshi population. Our study found a significant association of high rice intake with increased HCC risk. Previously, there was no evidence of association of rice intake with HCC risk; however, one study found null association between total carbohydrate intake and HCC risk [45]. As Bangladeshi people are more likely to eat rice frequently, it is relevant that rice may play a role as a dietary risk factor for HCC. Red meat has long been recognized as a dietary risk factor for HCC and significantly it has been proven that it has a positive relation with increased HCC risk. Huang et al. (2003) suggested that red meat intake may increase the risk of HCC. The polymorphism of the N-acetyltransferase 2 (NAT2) gene plays a role increasing the susceptibility of the effect of red meat in HCC development [46]. Cross et al. (2007) found that, red meat intake was associated with an elevated risk for liver cancer [47]. Red meat is the main source of heme iron and may increase HCC risk via the possible effect of iron such as hepatocyte injury and death and DNA damage in tissues by catalyzing lipid peroxidation [48]. In the current scenario, the red meat consumption in Bangladesh is limited due to its very high price and most poor people's intake of red meat is in moderate or low amounts. The amount of consumption and also the frequency of consumption may play an important role for being a significant dietary factor for HCC development. In the perspective of our study, red meat has no association with increased risk of HCC. In our study, higher egg consumption was found to have significance for increased risk of HCC. However, two Italian studies suggested an inverse association between higher egg consumption and increased HCC risk [10,49]. The study by Bamia et al. (2014) found an inverse association of tea intake with HCC risk, which was attributed to the presence of polyphenols as an antioxidant, especially in green tea [50]. Our study was unable to retrieve any significant association between tea intake and HCC risk. Roti is also consumed in Bangladesh as a primary source of carbohydrates. The present study found no significant association of roti intake and risk of HCC. Leafy vegetables are group of crop plants that are grown for their edible leaves. Examples of leafy vegetables are various types of spinach, cabbage, parsley, and lettuce. However, the distinction between leafy and non-leafy vegetables is not always clear. Non leafy vegetables are potato, cucumber, tomato, eggplant, cauliflower, etc. While investigating the association of vegetables, we classified them as the above mentioned two groups. Surprisingly, we found significant association of lower intake of leafy vegetables with higher HCC risk. In other words, an increased intake of leafy vegetables might decrease the risk of HCC. We found no significant association between non-leafy vegetables and HCC risk. A prospective cohort study in Japan showed inverse associations between the consumption of vegetables, green–yellow and green leafy vegetables and HCC risk [51]. An association of overall vegetable consumption with reduced HCC risk was also found in studies conducted in China [52], Serbia [53], Japan [54], and Italy [49]. It is also noteworthy that another study conducted in Greece found no association of vegetable intake with risk of HCC [55]. Our study found that higher consumption of fruits is significantly associated with decreased HCC risk. The role of fruit consumption in HCC risk is not properly established as there is substantial evidence of both association and no association. Talamini et al. (2006) and Vecchia et al. (1988) suggested a positive association of higher fruit intake with lower HCC risk [10,49]. In contrast, Yang et al. (2014) and Bamia et al. (2015) found no significant association [56,57]. The major fact is that vegetables and fruits are major sources of vitamins, minerals, antioxidants, and many bioactive compounds which are major effectors against cancer [44]. That's why vegetable and fruit intakes are more likely to contribute to the prevention of cancer. Vegetables and fruits are also a major source of dietary fiber. The

association of dietary fiber with decreased risk of HCC was found in a study conducted by Fedirko et al. (2012) [45]. In the case of fish and milk, we found that, increased consumption of these food items may reduce the risk of HCC. In other words, lower consumption of fish and milk have positive association with HCC risk. Higher consumption of white meat is evident in several studies to have a positive impact on reduced HCC risk [10,58,59] and no other studies have reported results dissimilar to these studies. Fish intake is also proven to have an inverse association with increased HCC risk in previous studies [60,61]. The supporting factor is that a diet rich in linoleic acid in foods such as white meats and fish, was inversely related to HCC risk [62]. White meat and fish have less saturated fat and are rich in polyunsaturated fatty acids (PUFA) [48]. HCC prevention is attributed to the function of n-3 PUFA, which possesses anti-inflammatory activity by inhibiting interleukin-1 and tumor necrosis factor [63]. Milk consumption is not widely studied. However, there is also supporting evidence of higher milk consumption with reduced HCC risk in a previous study [10].

From our study, it is evident that diet and food habits may play an essential role in the increased risk of HCC in the Bangladeshi population. Given cirrhosis is the major cause of HCC, the role of nutrition in the development of cirrhosis vs HCC warrants further research. In the perspective of our study, comparing the food habit pattern between cirrhosis positive and negative HCC patients did not give any significance due to the small number of valid responses (n = 80), and low number of responses per category per risk factor. We believe, more intense research is required to find out the exact mechanism for the association of food habits with the risk of HCC as an additional risk factor other than already established risk factors.

## Supporting information

**S1 Table. Bivariate association between food habit and hepatitis B or C positive against the others.**
(DOCX)

**S1 Questionnaire.**
(PDF)

**S2 Questionnaire.**
(PDF)

## Acknowledgments

We would like to thank the tertiary care facility, from where we got all our patients data.

## Author Contributions

**Conceptualization:** M. Rezaul Karim.

**Data curation:** M. Al-Amin Shawon, M. Abul Khair Yousuf, Tyeaba Tasnim Dipti.

**Formal analysis:** Enayetur Raheem.

**Funding acquisition:** M. Rezaul Karim.

**Investigation:** M. Abul Khair Yousuf, Sium Ahmed, Tyeaba Tasnim Dipti, M. Rezaul Karim.

**Methodology:** M. Al-Amin Shawon, M. Abul Khair Yousuf, Tyeaba Tasnim Dipti, M. Rezaul Karim.

**Project administration:** M. Rezaul Karim.

**Supervision:** Hiroaki Taniguchi, M. Rezaul Karim.

**Validation:** Mohammad Razuanul Hoque, Hiroaki Taniguchi.

**Visualization:** Mohammad Razuanul Hoque, Hiroaki Taniguchi.

**Writing – original draft:** M. Al-Amin Shawon, Sium Ahmed, M. Rezaul Karim.

**Writing – review & editing:** Enayetur Raheem, Mohammad Razuanul Hoque, Hiroaki Taniguchi, M. Rezaul Karim.

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
