## [Decision Letter · Decision Letter 0]

24 Feb 2020

PONE-D-19-34553

Epidemiology, clinical features, and impact of food habits on the risk of hepatocellular carcinoma: A case-control study in Bangladesh

PLOS ONE

Dear Associate Professor Karim,

Thank you for submitting your manuscript to PLOS ONE. Please accept our apologise if the revision process has taken longer than expected.

This editor and the reviewers have found this study intriguing and the results of value. However, both the reviewers have raised substantial concerns (that I share) regarding some of the conclusions and have suggested additional analyses and clarification before the manuscript can be considered for publication. Therefore we all recommended a mayor revision and we invite the authors to resubmit the manuscript if the authors are able to address the comments.

In specific, the reviewer 1 (expert of nutrition/epidemiology) has raised concerns regarding the methodology of the nutritional analysis (the food intake can be better dissected according to FFQ; coffee to be stratified according to intake), the criteria for recruitment (this is also raised by reviewer 2); the logistic regression strategy and the exposure categories should also better clarified; he also suggested re-writing some of the sentences in light of state of the art literature.

The reviewer 2 (expert of hepatology/HCC) has provided the authors with important suggestions regarding the methodology to carry the analyses keeping in consideration the progression of chronic liver disease (thich is the major determinant of HCC risk!): HCC is rarely observed in a normal liver and dissecting the mutual relationship between nutrition/chronic liver disease/HCC is crucial for a sound interpretation of the results. It needs to be clarified 1) the nature of the control group (real controls? Chronic hepatitis without HCC? the latter group would be extremely precious ...); 2) how the authors dissect the interaction between the smoking/drinking behaviour and the underlying chronic liver disease in their analyses (this might require some re-thinking on the strategy to analyse the data as suggested); 3) Analysing independently those patients with viral hepatitis with those with other etiologies: especially in patients with ASH/NASH, nutritional factors will be per se a driver of chronic liver disease progression (and not only a contributing risk factor); 4) dissect in a multivariate fashion the role of nutrition on the development of cirrhosis vs. HCC and, as a consequence, how nutritional habits influence the development of HCC in patients that do not have cirrhosis yet.

I also think that, since some metabolic information is available (BMI, T2D), these factors should be considered (together with chronic liver disease) in the multivariate approaches suggested: considering the growing concern of the impact that the obesity epidemics will have on HCC risk (not only because of NASH, but also as obesity is a worsening factor of chronic liver disease from different etiologies), and given obesity is pretty much associated to nutritional habits, these analyses will help to better dissect the relationship between nutrition/obesity/chronic liver disease thus leading to HCC.

My opinion is that these are all reasonable suggestions that will greatly improve the impact of this manuscript; the outcome of these analyses will also provide suggestions to the scientific community for further studies to address the role of nutrition on HCC development thus not limiting the importance to the study to the impact for Bangladeshi community.

English should be revised by a native English; there are also multiple spelling errors and typos.

We would appreciate receiving your revised manuscript by Apr 09 2020 11:59PM. To enhance the reproducibility of your results, we recommend that if applicable you deposit your laboratory protocols in protocols.io, where a protocol can be assigned its own identifier (DOI) such that it can be cited independently in the future. For instructions see: http://journals.plos.org/plosone/s/submission-guidelines#loc-laboratory-protocols

We look forward to receiving your revised manuscript.

Kind regards,

Michele Vacca, M.D., Ph.D.

Academic Editor

PLOS ONE

Journal Requirements:

3. Your ethics statement must appear in the Methods section of your manuscript. If your ethics statement is written in any section besides the Methods, please move it to the Methods section and delete it from any other section. Please also ensure that your ethics statement is included in your manuscript, as the ethics section of your online submission will not be published alongside your manuscript.

Reviewers' comments:

Reviewer's Responses to Questions

**Comments to the Author**

1. Is the manuscript technically sound, and do the data support the conclusions?

Reviewer #1: Yes

Reviewer #2: Partly

2. Has the statistical analysis been performed appropriately and rigorously? 

Reviewer #1: Yes

Reviewer #2: Yes

3. Have the authors made all data underlying the findings in their manuscript fully available?

Reviewer #1: Yes

Reviewer #2: Yes

4. Is the manuscript presented in an intelligible fashion and written in standard English?

Reviewer #1: Yes

Reviewer #2: Yes

5. Review Comments to the Author

Reviewer #1: Authors explored the association between food group intake in Bangladesh and risk of hepatocellular carcinoma (HCC) in a hospital-based case-control study.

Introduction is quite long, more attention should be payed rather to nutritional part.

Part from line 78-95 should be shortened.

Authors should provide an explanation why they did not consider the following food groups in their analysis: whole-grains and cereals, coffee, nuts and legumes. Authors should consider that moderate coffee intake has been associated with decreased risk of HCC (PMID: 28846640). I suggest that authors add analysis for this food groups as they were included in FFQ.

Introduction line 101-105 authors should rephrase the sentence and improve the cited bibliography as references 19-25 correspond motel to individual case-control studies (published 20 years ago!) rather than comprehensive studies. Authors should refer to recent umbrella reviews summarising knowledge regarding HCC from prospective cohort and case-control studies and providing the level of evidence, as well as the most recent meta-analysis. It should be:

“Different studies claimed inverse relation of milk (PMID: 31199182), fiber and whole-grains (PMID: 31964201), white meat (PMID: 24588342), fruits and vegetables (PMID: 30764679), and fish (PMID: 25534918) with HCC. On the contrary, high intake of red meat (PMID: 24588342) has been associated with increased risk of HCC, however, there is no sufficient evidence on the association between HCC and egg consumption (PMID: 31379223).”

Study population. It is necessary to specify city and country in which were enrolled the individuals. Design of the study should be clearly specified: hospital-based case-control study.

Table 5 present data from multiple logistic regression, please add all the variables used in the adjustment in the table’s footnote. Please present also unadjusted model of analysis. Adjustment covariates should be also listed in “statistical analysis section”.

I suggest authors revise English, as there are several misspellings along the manuscript.

It would be worth to provide exposure categories for example as g/day ml/day or serving/day.

Figures are fine.

Discussion seems fine.

Reviewer #2: The study is aimed to understand the present status of HCC in Bangladesh and more in detail to describe the role of different etiological factors in the development of HCC (viral agents, alcohol, smoking diabetes, food habits).

Main original elements are: the deep and appropriate analysis of food habits; the fact that it represents the first large study on this topic in Bangladeshi patients.

The collection of data is well carried out and the presentation of data ic clear and understandable.

Nevertheless, the study has some limitations and the authors should give further information:

1. Control subjects: it is not clear by which kind of people the control population is represented: are they normal subjects? are people affected by other tumors or other disease? are they represented by cirrhotic patients without HCC? This point is absolutely crucial in order to understand the results ot the table 1 of the work

2. Since smoking is considered a risk cofactor for tumors at all and for HCC (EASL, 2019), the author cannot state that the most common etiology for HCC was smoking (50% reporting) followed by the HBV infection. They need to better explain this concept, even because the so-called control group has a rather high percentage of smoking subjects (30%). Any consideration on the smoking as an etiological factor has to take into account the data that main etiological factor for HCC is the cirrhosis and the relationship between cirrhosis and smoking is not that clear.

3. What the authors mean for “alcohol consumption”? They have to better explain: is it intented as a general consideration on the possible role of alcohol consumption in the pathology of the patient or do they relate to a specific level of consumption?

Since the authors have carried out a very beautiful and detailed study on the food consumption, I would have expected some more details on this relevat etiological factor.

4. Surprisingly, 35 (44%) patients were discovered negative for both hepatitis B and hepatitis C infections: this is a very interesting data, but maybe the authors could analyse the data on food habits in this group of patients in comparison with the groups of patients with other established etiology (e.g. HBV and/or HCV). Their conclusions on the possible association between food habit and risk of HCC could be reinforced.

5. Are the authors able to distinguish between the role of food habits in the development of the underlying cirrhosis and the role in the comparison of HCC? This is not a peregrine observation, since cirrhosis is the most worldwide risk factor for HCC.

6. Actuallly, the authors need to clearly say how many HCC enclosed in this interesting work are to be considered primary or secondary to cirrhosis.

6. PLOS authors have the option to publish the peer review history of their article (what does this mean?). If published, this will include your full peer review and any attached files.

Reviewer #1: No

Reviewer #2: Yes: Vincenzo O. Palmieri

---

## [Author Response · Author response to Decision Letter 0]

26 Mar 2020

Dear Dr. Michele Vacca, M.D., Ph.D. 

We thank you and the reviewers for the generous comments on our manuscript and have edited the manuscript to address the proposed concerns. We include with this submission a Point by Point letter explaining the reviewers’ comments carefully. We note that both referees particularly commented on the high standard and thoroughness of our work. In this revised submission we have significantly extended and revised our work and have addressed both the general and specific comments of the referees. Therefore, we believe the manuscript is suitable for publication in Plos One.

Response to Editor’s Points:

It needs to be clarified 

1) The nature of the control group (real controls? Chronic hepatitis without HCC? The latter group would be extremely precious ...); 

Our response: The control group of this study is normal, healthy subjects (real control). 

2) How the authors dissect the interaction between the smoking/drinking behaviour and the underlying chronic liver disease in their analyses (this might require some re-thinking on the strategy to analyse the data as suggested); 

Our response: In our multivariate analysis, we found crude odds ratio for smoking to be significant (p=0.006) but the significance went away when adjusted for the rest of the predictors. This may suggest potential interaction between smoking behavior and one or more predictors in the model. This warrants further exploration.

Thus, we evaluated the effect of interactions between smoking and all the food intake variables. None of the interactions was significant when adjusted for the other predictors. None but one crude odds ratio was somewhat significant, which was leafyvegLow:smokingYes compared to leafyvegMedium:smokingNo.

Based on the findings, the interactions between smoking and food habit does not warrant to be included in the model.

We could not evaluate the interaction between sex and smoking because there was not a single female who smoke and have HCC. See the table below.

label levels Control HCC

sex F 20 (19.8) 16 (20.0)

 M 81 (80.2) 64 (80.0)

smoking No 71 (70.3) 40 (50.0)

 Yes 30 (29.7) 40 (50.0)

sex:smoking F|No 14 (13.9) 16 (20.0)

 F|Yes 6 (5.9) 

 M|No 57 (56.4) 24 (30.0)

 M|Yes 24 (23.8) 40 (50.0)

Since alcohol use is not as common in the study population, and also due to only a few responses who reported alcohol consumption, we did not investigate it’s interaction with food habit.

3) Analysing independently those patients with viral hepatitis with those with other etiologies: especially in patients with ASH/NASH, nutritional factors will be per se a driver of chronic liver disease progression (and not only a contributing risk factor); 

Our response: This is a great suggestion. However, our study only focused on analyzing HCC patients and we plan to analyze the suggested issues in the future studies. Thank you very much for your fruitful suggestion. 

4) Dissect in a multivariate fashion the role of nutrition on the development of cirrhosis vs. HCC and, as a consequence, how nutritional habits influence the development of HCC in patients that do not have cirrhosis yet.

Our response: We thank the editor for this insightful feedback. Based on our understanding of the feedback, we believe, this is a topic that warrants further research.

Yet, using the current data, we fitted a multivariable logistic regression model to predict the cirrhosis using the same set of risk factors used for modeling the HCC. We did not find anything significant. This is perhaps due to small valid responses (n=80), and low number of responses per category per risk factor. A future study with a larger sample size would be necessary to answer this question. (N.B. we have added few sentences, 587-591, in the discussion)

label levels HCC

Cirrhosis Indicator No 18 (22.5)

 Yes 62 (77.5)

## Warning in kable_markdown(x, padding = padding, ...): The table should have a

## header (column names)

Dependent: cirhosis_ind No Yes OR (univariable) OR (multivariable)

Age Mean (SD) 44.9 (16.3) 49.8 (14.3) 1.02 (0.99-1.06, p=0.216) 1.01 (0.97-1.06, p=0.588)

Sex F 6 (37.5) 10 (62.5) - -

 M 12 (18.8) 52 (81.2) 2.60 (0.76-8.52, p=0.116) 1.15 (0.22-5.96, p=0.862)

Smoker No 14 (35.0) 26 (65.0) - -

 Yes 4 (10.0) 36 (90.0) 4.85 (1.54-18.65, p=0.011) 3.92 (0.93-19.22, p=0.070)

Rice intake Moderate 2 (33.3) 4 (66.7) - -

 High 16 (21.6) 58 (78.4) 1.81 (0.24-10.19, p=0.514) 1.49 (0.06-24.49, p=0.786)

Egg intake Moderate 4 (13.3) 26 (86.7) - -

 High 14 (28.0) 36 (72.0) 0.40 (0.10-1.25, p=0.136) 0.32 (0.06-1.25, p=0.126)

Leafy vegetable intake Moderate 3 (12.0) 22 (88.0) - -

 Low 15 (27.3) 40 (72.7) 0.36 (0.08-1.25, p=0.140) 0.28 (0.05-1.21, p=0.115)

Fruit intake Moderate 1 (20.0) 4 (80.0) - -

 Low 17 (22.7) 58 (77.3) 0.85 (0.04-6.26, p=0.890) 0.55 (0.01-7.52, p=0.692)

Fish intake Moderate 5 (27.8) 13 (72.2) - -

 Low 13 (21.0) 49 (79.0) 1.45 (0.41-4.66, p=0.544) 1.00 (0.20-4.17, p=1.000)

Milk intake Moderate 5 (35.7) 9 (64.3) - -

 Low 13 (19.7) 53 (80.3) 2.26 (0.61-7.79, p=0.200) 3.34 (0.53-21.18, p=0.188)

White meat intake Moderate 8 (20.0) 32 (80.0) - -

 Low 10 (25.0) 30 (75.0) 0.75 (0.25-2.15, p=0.593) 0.75 (0.20-2.72, p=0.656)

Diabetes No 10 (17.5) 47 (82.5) - -

 Yes 8 (34.8) 15 (65.2) 0.40 (0.13-1.21, p=0.100) 0.91 (0.24-3.76, p=0.886)

Weight status Normal 11 (17.5) 52 (82.5) - -

 Overweight 7 (41.2) 10 (58.8) 0.30 (0.09-0.99, p=0.044) 0.37 (0.08-1.68, p=0.190)

Number in data frame = 181, Number in model = 80, Missing = 101, AIC = 93.2, C-statistic = 0.785, H&L = Chi-sq(8) 13.48 (p=0.096)

I also think that, since some metabolic information is available (BMI, T2D), these factors should be considered (together with chronic liver disease) in the multivariate approaches suggested: considering the growing concern of the impact that the obesity epidemics will have on HCC risk(not only because of NASH, but also as obesity is a worsening factor of chronic liver disease from different etiologies), and given obesity is pretty much associated to nutritional habits, these analyses will help to better dissect the relationship between nutrition/obesity/chronic liver disease thus leading to HCC.

Our response: We have included Type-2 diabetes and obesity status in the multivariable logistic regression model. However, they both do not appear to be significantly impacting the odds of HCC. The AIC=196.9 for the new model is slightly larger than the model without diabetes and weight status included in the model (AIC = 193.1) suggesting a slightly poorer fit. However, both models passed the Hosmer and Lemeshow goodness of fit test. (N.B. added in the discussion, 491-493)

Reviewers’ comments and our Point by Point response: 

Reviewer #1

Authors explored the association between food group intake in Bangladesh and risk of hepatocellular carcinoma (HCC) in a hospital-based case-control study.

-Introduction is quite long, more attention should be payed rather to nutritional part. Part from line 78-95 should be shortened.

Our response: We shortened the section, according to the reviewer. 

-Authors should provide an explanation why they did not consider the following food groups in their analysis: whole-grains and cereals, coffee, nuts and legumes. Authors should consider that moderate coffee intake has been associated with decreased risk of HCC (PMID: 28846640). I suggest that authors add analysis for this food groups as they were included in FFQ.

Our response: Whole-grains and cereals, coffee, and nuts are not very common in Bangladesh as a daily food habit. Legumes are consumed in Bangladesh as a vegetable, which we have included in our non-leafy vegetable group. 

While coffee intake is associated with lower risk of HCC, we found no correlation in the Bangladeshi community because of lack of coffee drinking. However, most people in Bangladesh drink tea, and we include the impact of tea in this study. 

-Introduction line 101-105 authors should rephrase the sentence and improve the cited bibliography as references 19-25 correspond motel to individual case-control studies (published 20 years ago!) rather than comprehensive studies. Authors should refer to recent umbrella reviews summarizing knowledge regarding HCC from prospective cohort and case-control studies and providing the level of evidence, as well as the most recent meta-analysis. It should be:

“Different studies claimed inverse relation of milk (PMID: 31199182), fiber and whole-grains (PMID: 31964201), white meat (PMID: 24588342), fruits and vegetables (PMID: 30764679), and fish (PMID: 25534918) with HCC. On the contrary, high intake of red meat (PMID: 24588342) has been associated with increased risk of HCC, however, there is no sufficient evidence on the association between HCC and egg consumption (PMID: 31379223).”

Our response: We are very grateful to the reviewer for the updated information and we also thank him for having been so helpful in editing the sentences. We changed the sentences as indicated by the reviewer.

-Study population. It is necessary to specify city and country in which were enrolled the individuals. Design of the study should be clearly specified: hospital-based case-control study.

Our response: We have added the following information in Material and methods, line 106 and 109. 

-Table 5 present data from multiple logistic regression, please add all the variables used in the adjustment in the table’s footnote. Please present also unadjusted model of analysis. Adjustment covariates should be also listed in “statistical analysis section”.

Our response: Table 5 (now Table 6, in the revised manuscript) only considered a subset of the original variables. We've now added a sentence to the Statistical Analysis section about our modeling strategies. There, we've mentioned that we first performed a bivariate analysis of all potential risk factors with the outcome of interest. Variables showing statistically significant at the 20% level, were considered for inclusion in the regression model. There were some exceptions to this rule. We've included age and sex regardless. Also, we've included diabetes and weight status (per the recommendation of one of the reviewers).

Since Table 5 (now Table 6) only lists the variables included in the model, there is no need to list them separately in the footnote. The table also lists crude odds ratio (univariate) as well as the adjusted odds ratio (multivariate setup adjusted for the remaining variables in the model).

We believe the approach we've taken is common in this type of study.

-I suggest authors revise English, as there are several misspellings along the manuscript.

Our response: We proofread the manuscript with the professional. 

-It would be worth to provide exposure categories for example as g/day ml/day or serving/day.

Our response: The measuring unit for solid food was g/day and for liquid was ml/day. We mentioned it in the materials and methods section, line 146-147.

Figures are fine.

Discussion seems fine.

Reviewer #2

The study is aimed to understand the present status of HCC in Bangladesh and more in detail to describe the role of different etiological factors in the development of HCC (viral agents, alcohol, smoking diabetes, food habits).

Main original elements are: the deep and appropriate analysis of food habits; the fact that it represents the first large study on this topic in Bangladeshi patients.

The collection of data is well carried out and the presentation of data ic clear and understandable.

Nevertheless, the study has some limitations and the authors should give further information:

-Control subjects: it is not clear by which kind of people the control population is represented: are they normal subjects? are people affected by other tumors or other disease? are they represented by cirrhotic patients without HCC? This point is absolutely crucial in order to understand the results of the table 1 of the work

Our response: The control population are healthy, normal subjects. And they are not affected by any tumor or other diseases. However, to avoid the artifact, during control selection we tried to match the age, sex, and socio-economic status of control subjects with the patient subjects. No control were cirrhotic patients. 

- Since smoking is considered a risk cofactor for tumors at all and for HCC (EASL, 2019), the author cannot state that the most common etiology for HCC was smoking (50% reporting) followed by the HBV infection. They need to better explain this concept, even because the so-called control group has a rather high percentage of smoking subjects (30%). Any consideration on the smoking as an etiological factor has to take into account the data that main etiological factor for HCC is the cirrhosis and the relationship between cirrhosis and smoking is not that clear. 

Our response: I agree with the reviewer that the main etiological factor for HCC is the cirrhosis. After analyzing our samples, we have found 62 out of 80 subjects were cirrhosis leading to HCC. Next we performed Fisher’s exact test for association between cirrhosis positive HCC patients and non-cirrhosis HCC patients. We find an odds ratio of 4.75 with 95% CI (1.3 to 22.1) and two-sided p=0.015. This suggests a four-fold odds of developing cirrhosis among the smokers than among the non-smokers.

- What the authors mean for “alcohol consumption”? They have to better explain: is it intended as a general consideration on the possible role of alcohol consumption in the pathology of the patient or do they relate to a specific level of consumption? Since the authors have carried out a very beautiful and detailed study on the food consumption, I would have expected some more details on this relevant etiological factor.

Our response: Only three subjects in this study have consumed alcohol. And they seldom drink alcohol, one or two times in a month. So we cannot conclude an association between alcohol and Bangladeshi HCC cases. 

- Surprisingly, 35 (44%) patients were discovered negative for both hepatitis B and hepatitis C infections: this is a very interesting data, but maybe the authors could analyse the data on food habits in this group of patients in comparison with the groups of patients with other established etiology (e.g. HBV and/or HCV). Their conclusions on the possible association between food habit and risk of HCC could be reinforced.

Our response: We compared patients with hepatitis B or C positive with the rest of the patients (categorized as ‘Other’) on food habits. We did not observe any statistically significant difference between the two groups on their food habits (Table S1).

- Are the authors able to distinguish between the role of food habits in the development of the underlying cirrhosis and the role in the comparison of HCC? This is not a peregrine observation, since cirrhosis is the most worldwide risk factor for HCC.

Our response: For this particular case-control study, we did not consider the cirrhosis positive control, hence we cannot conclude that food habits had a direct role in the development of the underlying cirrhosis.

- Actually, the authors need to clearly say how many HCC enclosed in this interesting work are to be considered primary or secondary to cirrhosis.

Our response: There were 80 HCC patients enclosed in this study. 62 out of 80 HCC patients were cirrhosis leading to HCC, while the rest, 18 HCC patients had no cirrhosis.

---

## [Decision Letter · Decision Letter 1]

8 Apr 2020

Epidemiology, clinical features, and impact of food habits on the risk of hepatocellular carcinoma: A case-control study in Bangladesh

PONE-D-19-34553R1

Dear Dr. Karim,

We are pleased to inform you that your manuscript has been judged scientifically suitable for publication and will be formally accepted for publication once it complies with all outstanding technical requirements.

With kind regards,

Michele Vacca, M.D., Ph.D.

Academic Editor

PLOS ONE

Additional Editor Comments (optional):

Dear Authors

all the reviewers and this editor were satisfied by your analyses and for addressing our queries. I am thus happy to let you know that the manuscript is now accepted for publication.

Well done

Michele Vacca

Reviewers' comments:

Reviewer's Responses to Questions

**Comments to the Author**

1. If the authors have adequately addressed your comments raised in a previous round of review and you feel that this manuscript is now acceptable for publication, you may indicate that here to bypass the “Comments to the Author” section, enter your conflict of interest statement in the “Confidential to Editor” section, and submit your "Accept" recommendation.

Reviewer #1: All comments have been addressed

Reviewer #2: All comments have been addressed

2. Is the manuscript technically sound, and do the data support the conclusions?

Reviewer #1: Yes

Reviewer #2: Yes

3. Has the statistical analysis been performed appropriately and rigorously? 

Reviewer #1: Yes

Reviewer #2: Yes

4. Have the authors made all data underlying the findings in their manuscript fully available?

Reviewer #1: Yes

Reviewer #2: Yes

5. Is the manuscript presented in an intelligible fashion and written in standard English?

Reviewer #1: Yes

Reviewer #2: Yes

6. Review Comments to the Author

Reviewer #1: Authors addressed all of the comments, and improved significantly the manuscript. I have no further comments.

Reviewer #2: The authors have widely satisflied the concerns I raised in my revision. I found very interesting their conclusion on the role of food habits, that is a very relevant issue in the study of pathogenesis of HCC.

7. PLOS authors have the option to publish the peer review history of their article (what does this mean?). If published, this will include your full peer review and any attached files.

Reviewer #1: No

Reviewer #2: Yes: Prof. Vincenzo O. Palmieri

---

## [Editor Report · Acceptance letter]

10 Apr 2020

PONE-D-19-34553R1 

Epidemiology, clinical features, and impact of food habits on the risk of hepatocellular carcinoma: A case-control study in Bangladesh 

Dear Dr. Karim:

I am pleased to inform you that your manuscript has been deemed suitable for publication in PLOS ONE. Congratulations! Your manuscript is now with our production department. 

With kind regards,

on behalf of

Dr. Michele Vacca 

Academic Editor

PLOS ONE